
# Diagonal fields in critical loop models

**Sylvain Ribault**

Université Paris-Saclay, CNRS, CEA, Institut de physique théorique,
91191, Gif-sur-Yvette, France

sylvain.ribault@ipht.fr

## Abstract

In critical loop models, there exist diagonal fields with arbitrary conformal dimensions, whose 3-point functions coincide with those of Liouville theory at $c \leq 1$. We study their $N$-point functions, which depend on the $2^{N-1}$ weights of topologically inequivalent loops on a sphere with $N$ punctures. Using a numerical conformal bootstrap approach, we find that 4-point functions decompose into infinite but discrete linear combinations of conformal blocks. We conclude that diagonal fields belong to an extension of the $O(n)$ model.

## 1   Predictions from loop models

In two dimensions, statistical models such as the $O(n)$ model and the Potts model can be described using ensembles of non-intersecting loops on a lattice. These models have a critical limit where they become conformally invariant. Although the lattice disappears in that limit, it is still possible to make sense of the loops using conformal loop ensembles. Here we will investigate whether the critical limit can be interpreted as a conformal field theory. This would allow

observables to be computed to high precision, or even exactly, using the conformal bootstrap method. We will focus on correlation functions of diagonal fields, a class of observables that are simple to define, and nevertheless encode much information through their dependence on several continuous variables.

We focus on a loop model on a sphere with $N$ punctures $z_i \in \mathbb{C} \cup \{\infty\}$, where the loops avoid the punctures. Each loop $\mathcal{C}$ gives rise to a 2-partition $P(\mathcal{C})$ of the punctures, i.e. a partition into two (possibly empty) sets, for example:

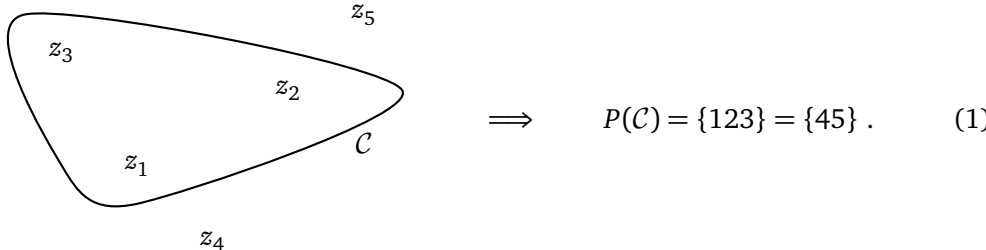

$$\implies \qquad P(\mathcal{C}) = \{123\} = \{45\} . \tag{1}$$

Since the 2-partition is a topological property of the loop, the loop's weight can depend on the 2-partition without spoiling conformal invariance. The model's partition function takes the form

$$Z_N^{\text{lattice}}(w, \{z_i\}) = \sum_{E \in \mathcal{E}} \prod_{\mathcal{C} \in E} w(P(\mathcal{C})) , \tag{2}$$

where $\mathcal{E}$ is an ensemble of configurations of non-intersecting loops, and the loop weight $w \in \mathbb{C}^{2^{N-1}}$ is a function on the set of 2-partitions.

In the critical limit, we would like to interpret the partition function as a correlation function of $N$ primary fields,

$$\lim_{\text{critical}} Z_N^{\text{lattice}}(w, \{z_i\}) = \left\langle \prod_{i=1}^{N} V_{\Delta_i}(z_i) \right\rangle . \tag{3}$$

Here $V_\Delta(z)$ is a diagonal primary field, whose left and right conformal dimensions are both $\Delta$. The central charge and conformal dimensions are related to loop weights by [1]

$$w(\emptyset) = -2\cos(\pi\beta^2) \qquad \text{with} \qquad c = 13 - 6\beta^2 - 6\beta^{-2} , \tag{4}$$

$$w(\{i\}) = 2\cos(2\pi\beta P_i) \qquad \text{with} \qquad \Delta_i = \frac{c-1}{24} + P_i^2 . \tag{5}$$

For $N = 1$ we have $w(\emptyset) = w(\{1\})$ and for $N = 2$ we have $w(\{1\}) = w(\{2\})$: these constraints are compatible with constraints on $\Delta_i$ from conformal symmetry, respectively $\Delta_1 = 0$ and $\Delta_1 = \Delta_2$. For $N = 3$, the 4 possible loop weights match the 4 parameters $c, \Delta_1, \Delta_2, \Delta_3$, and the critical limit of the lattice partition function was found to agree with a three-point function in Liouville theory with $c \leq 1$ [1], which also agrees with a three-point function in a conformal loop ensemble [2]:

$$\lim_{\text{critical}} Z_3^{\text{lattice}}(w, \{z_i\}) = \left\langle \prod_{i=1}^{3} V_{\Delta_i}(z_i) \right\rangle_{c \leq 1 \text{ Liouville theory}} = Z_3^{\text{CLE}_c}(\Delta_i, z_i) . \tag{6}$$

For $N = 4$, the 8 possible loop weights correspond to the 5 parameters $c, \Delta_1, \Delta_2, \Delta_3, \Delta_4$, plus the three weights $w(\{12\}), w(\{14\}), w(\{13\})$. We call $P_s, P_t, P_u \bmod \beta^{-1}\mathbb{Z}$ the corresponding momenta via Eq. (5), for example $w(\{12\}) = 2\cos(2\pi\beta P_s)$. We write

$Z_4(P_s, P_t, P_u) = \lim_{\text{critical}} Z_4^{\text{lattice}}(w, \{z_i\})$ our four-point function. Since the loops are non-intersecting, loops with $P(\mathcal{C}) = \{12\}$ cannot coexist with loops with $P(\mathcal{C}) = \{13\}$ or $P(\mathcal{C}) = \{14\}$:

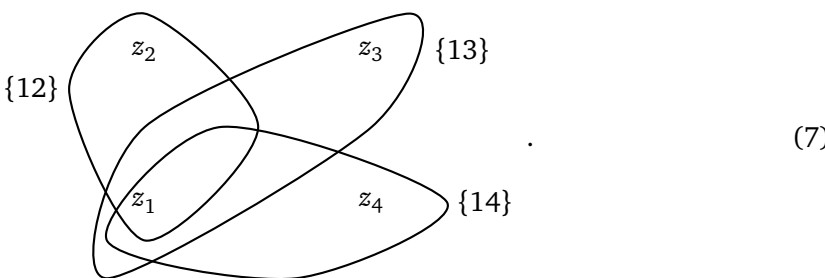

$$\tag{7}$$

Therefore, the ensemble $\mathcal{E}$ of loop configurations splits into four subsets, depending on the existence of such loops:

$$\mathcal{E} = \mathcal{E}_{\{12\}} \sqcup \mathcal{E}_{\{13\}} \sqcup \mathcal{E}_{\{14\}} \sqcup \left\{ E \in \mathcal{E} \, \big| \, P(E) \subset \{\emptyset, \{1\}, \{2\}, \{3\}, \{4\}\} \right\} , \tag{8}$$

where $\mathcal{E}_{\{12\}} = \left\{ E \in \mathcal{E} \, \big| \, \{12\} \in P(E) \right\}$. The lattice sum (2) therefore splits into four terms, and the sum over $\mathcal{E}_{\{12\}}$ is the only term that depends on $w(\{12\})$. In the critical limit, the four-point function $Z_4(P_s, P_t, P_u)$ therefore obeys linear relations of the type $\partial_{P_s} \partial_{P_t} Z_4 = 0$, or equivalently

$$Z_4(P_s, P_t, P_u) + Z_4(P'_s, P'_t, P_u) = Z_4(P'_s, P_t, P_u) + Z_4(P_s, P'_t, P_u) , \tag{9}$$

for any choice of $P_s, P_t, P_u, P'_s, P'_t$.

It is already clear that $Z_4(P_s, P_t, P_u)$ is not a four-point function in Liouville theory, because a Liouville four-point function would not depend on $P_s, P_t, P_u$. Moreover, in contrast to three-point functions, Liouville four-point functions are not analytic as functions of the central charge, due to non-analyticities on the half-line $\{c \leq 1\}$ [3]. But the lattice sum $Z_N^{\text{lattice}}(w, \{z_i\})$ is manifestly analytic in $c$, and there is no hint that the critical limit produces singularities on the half-line $\{c \leq 1\}$ [4]. We will therefore propose a construction of the four-point function $Z_4(P_s, P_t, P_u)$ that is not based on Liouville theory.

## 2 Ansatz for the four-point function

In order to compute a four-point function of the type (3) with the semi-analytic bootstrap methods of [5], we need to specify the spectrum, i.e. the set of fields that propagate in each channel. In the case of $Z_4(P_s, P_t, P_u)$, we make three assumptions:

1. In each channel $x \in \{s, t, u\}$, there are diagonal primary fields with momentums in $P_x + \beta^{-1}\mathbb{Z}$.

2. The rest of the primary fields are $V^N_{(r,s)}$ with $r \in \mathbb{N}^*$ and $s \in \frac{1}{r}\mathbb{Z}$. Here we define $V^N_{(r,s)}$ to have left and right dimensions $(\Delta, \bar{\Delta}) = (\Delta_{(r,s)}, \bar{\Delta}_{(r,-s)})$, with

$$\Delta_{(r,s)} = \tfrac{1}{4}(r\beta - s\beta^{-1})^2 - \tfrac{1}{4}(\beta - \beta^{-1})^2 . \tag{10}$$

3. The behaviour of structure constants under $P_x \to P_x + \beta^{-1}$ or $s \to s + 2$ is determined by the shift equations that follow from the existence of the degenerate diagonal field $V_{(1,3)}$.

The second assumption amounts to including all fields in the $O(n)$ model that are allowed by fusion rules. This is reasonable because the $O(n)$ model is the known critical limit of a loop model, so we might as well use its space of fields [6, 7]. The $O(n)$ model fields that are

forbidden by fusion rules are degenerate fields, whose presence would imply relations between $P_1, P_2, P_3, P_4$, and fields with $r \in \frac{1}{2}\mathbb{N}^*$, which would violate the conservation of $r \bmod \mathbb{Z}$, where by convention $r = 0$ for a diagonal field. Nevertheless, with our third assumption, we retain constraints that follow from the existence of correlation functions of the type $\langle V_{\langle 1,3\rangle} \cdots \rangle$ [8].

Our three assumptions lead to the following ansatz for the four-point function:

$$\forall x \in \{s, t, u\}\,, \quad Z_4(P_s, P_t, P_u) = D_{P_x}^{(x)} \mathcal{G}_{P_x}^{(x)} + \sum_{r \in \mathbb{N}^*} \sum_{\substack{s \in \frac{1}{r}\mathbb{Z} \\ -1 < s \leq 1}} D_{(r,s)}^{(x)} \mathcal{G}_{(r,s)}^{(x)}\,, \tag{11}$$

where $D_k^{(x)}$ are unknown structure constants (i.e. they are $z_i$-independent), and $\mathcal{G}_k^{(x)}$ are known $x$-channel interchiral blocks. The interchiral blocks are infinite linear combinations of conformal blocks, obtained by summing over $P_x + \beta^{-1}\mathbb{Z}$ or $s + 2\mathbb{Z}$ [9]. The relevant conformal blocks are products of left- and right-moving Virasoro blocks, except in the case of $\mathcal{G}_{(r,s)}^{(x)}$ with $s \in \{0, 1\}$, which involves the non-factorizable logarithmic conformal blocks that have been determined in [10].

## 3 Numerical bootstrap results

Our ansatz (11) may be viewed as a system of linear equations for the structure constants. Building on existing bootstrap code, we have solved these equations numerically [11]. We have found three main results:

1. After fixing the normalization by setting the value of one structure constant (say $D_{P_s}^{(s)}$), the solution of the system is unique. Numerically, this means that the solutions of truncated systems of finitely many equations with finitely many unknowns converge when the truncation parameter becomes large. This result shows that our ansatz does lead to a value for $Z_4(P_s, P_t, P_u)$, which can be computed numerically to any given precision.

2. There exists a normalization such that the structure constants for the diagonal fields factorize as

$$D_{P_s}^{(s)} = \frac{C_{P_1, P_2, P_s} C_{P_3, P_4, P_s}}{B_{P_s}}\,, \quad D_{P_t}^{(t)} = \frac{C_{P_1, P_4, P_t} C_{P_2, P_3, P_t}}{B_{P_t}}\,, \quad D_{P_u}^{(u)} = \frac{C_{P_1, P_3, P_u} C_{P_2, P_4, P_u}}{B_{P_u}}\,, \tag{12}$$

where $B, C$ are two- and three-point structure constants of Liouville theory with $c \leq 1$, analytically continued to complex values of $c$.

3. The resulting four-point function obeys the linear relation (9).

Let us illustrate the first two results in a numerical example. We have chosen the following values of the parameters, with $i = \sqrt{-1}$:

$$\beta = \frac{10}{8+i}\,, \quad (P_1, P_2, P_3, P_4) = \frac{1}{20\beta}\left(1 + 5i, 2 + 2i, 3 + 6i, 4 + i\right)\,, \tag{13}$$

$$\left(P_s^{-1}, P_t^{-1}, P_u^{-1}\right) = 2\beta\left(3 + i, 4 + i, 5 + i\right)\,. \tag{14}$$

The numerical calculations are performed with 32 decimal digits, and the spectrums are truncated to the maximal conformal dimension $\Delta + \bar{\Delta} = 40$. The structure constant $D_{P_s}^{(s)}$ is normalized in terms of Liouville theory structure constants as in Eq. (12). Let us then display the resulting values of the first few $t$-channel structure constants (in real part), together with

their deviations (i.e. relative numerical accuracies):

| Constant | Value | Deviation |
|---|---:|---|
| $D^{(t)}_{P_t}$ | 1.05304088130261122441863359665327 | $2.4 \times 10^{-26}$ |
| $D^{(t)}_{(1,0)}$ | −0.077726954973226305419163676742536 | $5.5 \times 10^{-26}$ |
| $D^{(t)}_{(1,1)}$ | −0.02292944151804650320517424817961 | $1.5 \times 10^{-25}$ |
| $D^{(t)}_{(2,0)}$ | 0.00002425464068749936811259321062107 | $5.1 \times 10^{-23}$ |
| $D^{(t)}_{(2,\frac{1}{2})}$ | 0.000048810625175510399035535766255115 | $1.3 \times 10^{-22}$ |
| $D^{(t)}_{(2,1)}$ | −0.000038875104437278368651041889030702 | $6.3 \times 10^{-23}$ |
| $D^{(t)}_{(2,-\frac{1}{2})}$ | 0.000048810625175510399035508778511608 | $2.7 \times 10^{-23}$ |
| $D^{(t)}_{(3,0)}$ | −0.0000000004067330030862373393865 | $4.6 \times 10^{-14}$ |

(15)

In particular, this can be compared with the analytic prediction from (12)

$$\Re D^{(t)}_{P_t} \simeq \underline{1.05304088130261122442}03458610973 \; . \tag{16}$$

This agrees with the numerical result to 20 significant digits (underlined). The bootstrap result itself is supposed to be accurate to about 26 digits, according to its deviation, but the calculation based on the analytic formula is a bit less accurate. If we increase the numerical cutoffs, the agreement improves, and the deviations decrease, which signals convergence towards a solution that obeys Eq. (12) [8].

This is strong evidence that our ansatz does describe a four-point function in the critical limit of the loop model. In particular, the four-point structure constants $D^{(x)}_{P_x}$ factorize into three-point structure constants that were directly compared to lattice results in [1]. Of course, it would be interesting to directly compare our four-point function with lattice sums. This could also help explain why the lattice sums are periodic in the momentums $P_i$ (see Eq. (5)) whereas $Z_4(P_s, P_t, P_u)$ is not. In the case of the dependence on the central charge (4), this apparent discrepancy is explained by the loss of criticality of the lattice model outside a restricted region in $c$-space [4]: does a similar phenomenon occur for the dependence on $P_i$?

## 4 Factorization of structure constants

In a consistent conformal field theory, the structure constants $D^{(x)}_{(r,s)}$ should decompose into two- and three-point structure constants, just like $D^{(x)}_{P_x}$. The decomposition can have several terms, depending on the multiplicity $m_{(r,s)}$ of the field $V^N_{(r,s)}$:

$$D^{(s)}_{(r,s)} \overset{?}{=} \sum_{k=1}^{m_{(r,s)}} \frac{C^{[k]}_{P_1,P_2,(r,s)} C^{[k]}_{P_3,P_4,(r,s)}}{B^{[k]}_{(r,s)}} \; . \tag{17}$$

In the $O(n)$ model, these multiplicities may be described in terms of representations of the global symmetry group $O(n)$. However, our diagonal fields do not belong to the $O(n)$ model, and they presumably break $O(n)$ symmetry. At least, diagonal fields cannot be $O(n)$ singlets, since many fields of the type $V^N_{(r,s)}$ cannot transform as singlets, starting with $V^N_{(1,0)}$ (symmetric two-tensor) and $V^N_{(1,1)}$ (antisymmetric two-tensor) [7].

So we lack an a priori understanding of the multiplicities. Worse, we find that the structure constants $D_{(r,s)}^{(s)}$ depend on $P_s, P_t, P_u$. So the decomposition (17) cannot hold. Nevertheless, from the linear equation (9), we expect that each structure constant is a sum of terms, with each term depending on only one momentum $P_s, P_t$ or $P_u$. We have numerically investigated how the dependence in the momentums factorizes in each term. Schematically, we find

$$D_{(r,s)}^{(s)}(P_s, P_t, P_u) = \sum_{k=1}^{m_{(r,s)}} C_{P_1,P_2,(r,s)}^{[k]} C_{P_3,P_4,(r,s)}^{[k]} \left( \tilde{B}_{(r,s)}^{[k]} + \sum_{x \in \{s,t,u\}} \tilde{B}_{(r,s)}^{[k],x}(P_x) \right). \qquad (18)$$

In other words, the dependences on $P_1, P_2$, on $P_3, P_4$ and on $P_x$ factorize. We use the notation $\tilde{B}_{(r,s)}^{[k],x}(P_x)$ rather than $\frac{1}{B_{(r,s)}^{[k],x}(P_x)}$ because this term vanishes in some cases, and is not really an inverse two-point structure constant since it depends on $P_x$. Our numerical methods actually give us access to the first few values of $(r,s)$, for which we find the following numbers of terms:

$$\begin{array}{c|cc|ccc|cccc} (r,s) & (1,0) & (1,1) & (2,0) & (2,\frac{1}{2}) & (2,1) & (3,0) & (3,\frac{1}{3}) & (3,\frac{2}{3}) & (3,1) \\ \hline m_{(r,s)} & 2 & 2 & 3 & 2 & 3 & 4 & 3 & 3 & 4 \end{array}. \qquad (19)$$

(We only display values of $s$ in the interval $0 \leq s \leq 1$, because $m_{(r,s)} = m_{(r,-s)} = m_{(r,s+2)}$, due to the shift equations for structure constants.) Let us briefly indicate how we determine these numbers. A function of two variables factorizes into $m$ terms $f(x,y) = \sum_{k=1}^{m} f^{[k]}(x)g^{[k]}(y)$ if and only if for any values $x_1, x_2, \ldots, x_{m+1}$ and $y_1, y_2, \ldots, y_{m+1}$ we have $\det[f(x_k, y_\ell)]_{1 \leq k, \ell \leq m+1} = 0$. We therefore compute our four-point structure constants $D_{(r,s)}^{(x)}(P_s, P_t, P_u)$ for various values of the momenta $P_i, P_x$, and deduce the corresponding determinants. Numerically, these determinants are never exactly zero, but in practice we consider that $\det F \simeq 0 \iff \forall k, \ell, \left| F_{k,\ell} F_{k,\ell}^{-1} \right| \gg 1$.

## 5 Relations with the $O(n)$ model and Liouville theory

Let us discuss the interpretation of the four-point functions $Z_4(P_s, P_t, P_u)$ in terms of conformal field theory. We start with the dependence on the central charge. Actually, our four-point functions are not functions of $c$, but of $\beta^2$ (4), i.e. they are not invariant under $\beta \to \beta^{-1}$. This is because our ansatz (11) is itself not invariant, since $\Delta_{(r,s)}$ (10) is not. On the other hand, Liouville theory is invariant, and so are the three-point structure constants $C_{P_1,P_2,P_3}$. Our four-point functions are defined over the same space $\{\Re \beta^2 > 0\}$ as the $O(n)$ model, which corresponds to a double cover of the space $\{\Re c < 13\}$. Over this space, we conjecture that our four-point functions depend analytically on $c$, because the conformal blocks do, and the sum in our ansatz is discrete and convergent. (Actually, a conformal block can have a pole for some $\beta^2 \in \mathbb{Q}$, but then the residue is another block, and the four-point function itself can be smooth thanks to the cancellation of singularities between different terms [12, 13].) Our numerical results are consistent with this conjecture: we find the that structure constants $D_{(r,s)}^{(x)}$ decrease quickly as $r$ increases, and we do not observe numerical instabilities that would betray the presence of singularities.

In contrast, Liouville theory is invariant under $\beta \to \beta^{-1}$. Moreover, Liouville four-point functions have an essential singularity over the whole half-line $\{c \leq 1\}$, due to the integration over a continuous spectrum [3]. If we wanted to compute Liouville four-point functions as sums over loops, we could certainly not use our straightforward construction with the simple loop weights (5). A more complicated construction is known to describe four-point functions in minimal models [14], and Liouville four-point functions on $\{c \leq 1\}$ should follow by taking limits in the central charge and conformal dimensions.

If our four-point functions have nothing to do with Liouville theory, why do they involve the Liouville structure constants $C_{P_1,P_2,P_3}$? And why do such structure constants describe three-point functions [1]? This may be a manifestation of universality. The structure constants $C_{P_1,P_2,P_3}$, although they first appeared in the context of Liouville theory with $c \leq 1$, are in fact unique solutions of certain shift equations [15, 16]. Such shift equations follow from the existence of two independent degenerate fields $V_{\langle 1,2 \rangle}$ and $V_{\langle 2,1 \rangle}$. The $O(n)$ model has the degenerate field $V_{\langle 1,3 \rangle}$, which leads to the same shift equation as $V_{\langle 1,2 \rangle}$ [8]. We have used these shift equations when computing interchiral blocks in our ansatz (11). But the $O(n)$ model does not include $V_{\langle 2,1 \rangle}$, and the real problem is to understand why some structure constants nevertheless obey the corresponding shift equations. The same problem arises in the Potts model, where the three-point connectivity is a special case of $C_{P_1,P_2,P_3}$ [10,17]. To summarize:

| Model | Liouville theory with $c \leq 1$. | (Extended) $O(n)$ model. |
|---|---|---|
| Parameter | $c \in (-\infty, 1]$ | $\beta^2 \in \left\{ \Re \beta^2 > 0 \right\}$ |
| Degenerate fields | $V_{\langle 2,1 \rangle}$ , $V_{\langle 1,2 \rangle}$ | $V_{\langle 1,3 \rangle}$ |
| Spectrum | Continuous, diagonal. | Discrete, non-diagonal. |

$$(20)$$

Therefore, the $N$-point functions $Z_N$ belong to an extension of the $O(n)$ model. This extension includes diagonal fields with arbitrary conformal dimensions, in the sense that their correlation functions exist, including correlation functions $\left\langle \prod_i V_{\Delta_i} \prod_j V^N_{(r_j,s_j)} \right\rangle$ that mix diagonal fields with non-diagonal $O(n)$ fields. However, the spectrum remains discrete: decompositions of correlation functions into conformal blocks are sums, not integrals. The extension may well be large enough for also including the Potts model, since there exist consistent four-point functions that mix fields from the $O(n)$ and Potts models [18]. In fact, the diagonal field with the conformal dimension $\Delta_{(0,\frac{1}{2})}$ plays a special role both in the Potts model, where it describes connectivities, and in the loop model, where the corresponding loop weight is $w = 0$. And in our numerical bootstrap calculations, we find that the structure constants $D^{(s)}_{(r,s)}(P_s, P_{(0,\frac{1}{2})}, P_{(0,\frac{1}{2})})$ decompose into fewer terms than in the generic case $D^{(s)}_{(r,s)}(P_s, P_t, P_u)$ (19), and also fewer terms than in the special case $D^{(s)}_{(r,s)}(P_s, P_t, P_t)$. We however do not have a precise explanation for these suggestive observations.

While we can compute its correlation functions, it is hard to make sense of our extension of the $O(n)$ model as a consistent conformal field theory, because of the dependence of $Z_4(P_s, P_t, P_u)$ on the momentums $P_s, P_t, P_u$. Due to this dependence, there is no factorization of the type (17), and this seems to violate the existence of the operator product expansion. Possible interpretations include:

1. We might interpret the observed factorization (18) in terms of a non-local operator product expansion, which would depend on $P_s, P_t, P_u$. This would add to the difficulty of defining and computing operator products in loop models — a tricky subject to begin with [19].

2. We might interpret $Z_4(P_s, P_t, P_u)$ as a four-point function in the presence of three topological defects with parameters $P_s, P_t, P_u$. In contrast to the loops themselves, the defects would intersect as in Figure (7). The four-point function without defects would then be $Z_4(P_{(0,\frac{1}{2})}, P_{(0,\frac{1}{2})}, P_{(0,\frac{1}{2})})$.

## Acknowledgements

I am grateful to Linnea Grans-Samuelsson, Jesper Jacobsen, Rongvoram Nivesvivat and Hubert Saleur for stimulating collaboration on closely related subjects. I wish to thank Hubert Saleur for comments and suggestions on the manuscript. I am grateful to Ingo Runkel and Xin Sun for helpful correspondence, and suggestions on the manuscript. I wish to thank the anonymous SciPost referee for suggestions that led to important clarifications.

This work is partly a result of the project ReNewQuantum, which received funding from the European Research Council.

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
