# Peer review of "Diagonal fields in critical loop models"

_SciPost Physics Core, doi:SciPost Phys. Core 6, 020 (2023)_

## Round 1 · Referee Report · Anonymous · 2022-12-7

Strengths
1- interesting subject
2-new application of bootstrap approach
Weaknesses
1- the presentation of the main results is very abstract
2- references are minimal
Report
The author studied partition function of ensembles of loops on punctured sphere which could be interpreted correlation functions in a CFT. By making some assumptions on the spectrum, the author then bootstrap to solve some 4 point functions with varying parameters. The results includes curious solutions such as the diagonal fields conformal data given by that of the time-like Liouville theory. The author then conclude that corresponding $N$-point function belong to an extension of $O(n)$ CFT.
While the subject is interesting, and the results are curious, I find some of the presentation not completely clear. I propose the following changes to improve the quality of the paper.
Requested changes
1- the author claimed in table (15) that the CFT under investigation involves degenerate field $V_{\langle1,3\rangle}$ (although this is not part of the spectrum). Did the author use the interchiral block resulting from such degeneracy in the boostrap? This is not clearly stated in the paper.
2- the bootstrap was carried out by choosing a series of parameters $P_s, P_t, P_u$. What are the choices used for obtaining the results?
3- result 1 claims that the system has a one-dimensional space of solutions for a given choice of $P_s, P_t, P_u$. How is this one-dimensional space parametrized? The author could plot some conformal data for example to illustrate this.
4- above table (15) the author claims that structure constants in the Potts model obey shift equations, despite lacking the degenerate field. I believe this point is not true. In 2005.07258, it was obtained that the Potts structure constants obey a modified version of the shift relation (dubbed renormalized Liouville recursion there) rather than that results from the degeneracy of $V_{\langle2,1\rangle}$.

---

## Round 2 · Referee Report · Anonymous (Referee 1) · 2022-12-15

Report

The author has addressed the suggestions of changes I made in the previous report.

It is interesting that once the overall normalization is fixed, there is a unique solution to the bootstrap problem under the given assumption. It seems to me that one interesting question would be to make a physical choice of the normalization (perhaps the particular solution of (12) with proper physical interpretation) and study how the structure constants for the non-diagonal fields in this case are related to that of the $O(n)$ model. This however may require an extensive amount of numerical work and could be left for future work.

I recommend the paper for publication.
  • validity: -
  • significance: -
  • originality: -
  • clarity: -
  • formatting: -
  • grammar: -

Author:  Sylvain Ribault  on 2022-12-15  [id 3139]

(in reply to Report 1 on 2022-12-15)
Category:
suggestion for further work

The referee's suggestions are spot on. Yes, (12) is surely the "correct" normalization, although its interpretation from the lattice is not too clear. And yes, studying structure constants of non-diagonal fields is very interesting, but requires quite a lot of extra work.

---

## Round 2 · Author Response

I have tried to clarify the text, following the referee's suggestions.

---

## Round 2 · List of Changes

1. I have made the use of interchiral symmetry more explicit, by replacing conformal blocks with interchiral blocks in the ansatz (10). I have also added explanations, including the new reference [9]. And interchiral symmetry now appears as a third assumption for the four-point functions $Z_4(P_s,P_t,P_u)$.

  2. I have added a numerical example, with parameter values in (12), (13), and results in (14).

  3. I have made the solution unique by fixing the normalization.

  4. I removed the claim about shift equations in the Potts model. The claim could have been made more precise, by explaining that the shift equations have extra factors compared to what we would expect from a degenerate field. The existence of these modified shift equations surely begs for an explanation. However, for the present paper's purpose, it is probably enough to mention the fact that the three-point connectivity coincides with a Liouville structure constant.

There are other small changes and clarifications. There used to be three sections, there are now five: the section that was called "Bootstrap results for four-point functions" is now split into three sections. With so many sections, it became more reasonable to have a table of contents.

---

## Editorial Decision

published